

# Antiviral drug discovery by targeting the SARS-CoV-2 polyprotein processing by inhibition of the main protease

Mahmoud Kandeel[1,2], Jinsoo Kim[3], Mahmoud Fayez[4,5], Yukio Kitade[6] and Hyung-Joo Kwon[3]

[1] Department of Biomedical Sciences, College of Veterinary Medicine, King Faisal University, Al-hofuf, Al-ahsa, Saudi Arabia
[2] Department of Pharmacology, Faculty of Veterinary Medicine, Kafrelshikh University, Kafrelshikh, Egypt
[3] Department of Microbiology, College of Medicine, Hallym University, Chuncheon, South Korea
[4] Al-Ahsa Veterinary Diagnostic Laboratory, Ministry of Agriculture, Al-Ahsa, Saudi Arabia
[5] Veterinary Serum and Vaccine Institute, Cairo, Dokki, Egypt
[6] Department of Applied Chemistry, Faculty of Engineering, Aichi Institute of Technology, Toyota, Japan

Corresponding authors
Mahmoud Kandeel,
mkandeel@kfu.edu.sa
Hyung-Joo Kwon,
hjookwon@hallym.ac.kr

## ABSTRACT

The spread of SARS-CoV-2, the causative agent for COVID-19, has led to a global and deadly pandemic. To date, few drugs have been approved for treating SARS-CoV-2 infections. In this study, a structure-based approach was adopted using the SARS-CoV-2 main protease (M$^{pro}$) and a carefully selected dataset of 37,060 compounds comprising M$^{pro}$ and antiviral protein-specific libraries. The compounds passed two-step docking filtration, starting with standard precision (SP) followed by extra precision (XP) runs. Fourteen compounds with the highest XP docking scores were examined by 20 ns molecular dynamics simulations (MDs). Based on backbone route mean square deviations (RMSD) and molecular mechanics/generalized Born surface area (MM/GBSA) binding energy, four drugs were selected for comprehensive MDs analysis at 100 ns. Results indicated that birinapant, atazanavir, and ritonavir potently bound and stabilized SARS-CoV-2 M$^{pro}$ structure. Binding energies higher than −102 kcal/mol, RMSD values <0.22 nm, formation of several hydrogen bonds with M$^{pro}$, favourable electrostatic contributions, and low radii of gyration were among the estimated factors contributing to the strength of the binding of these three compounds with M$^{pro}$. The top two compounds, atazanavir and birinapant, were tested for their ability to prevent SARS-CoV-2 plaque formation. At 10 µM of birinapant concentration, antiviral tests against SARS-CoV-2 demonstrated a 37% reduction of virus multiplication. Antiviral assays demonstrated that birinapant has high anti-SARS-CoV-2 activity in the low micromolar range, with an IC50 value of 18 ± 3.6 µM. Therefore, birinapant is a candidate for further investigation to determine whether it is a feasible therapy option.

## INTRODUCTION

The recent outbreak of SARS-CoV-2 has been declared a pandemic by WHO (*Cucinotta & Vanelli, 2020*). The disease COVID-19 causes a range of symptoms, from mild respiratory symptoms to severe respiratory distress associated with sepsis, multi-organ dysfunction, and

death (*Zaim et al., 2020*). The current alarming situation necessitates the rapid reallocation or repurposing of previously known drugs or chemical compounds for the use in treating COVID-19.

Approximately seven human coronaviruses (HCoV) have been identified. Four CoVs were identified as causative agents for mild respiratory symptoms and the common cold, including HCoV-229E, HCoV-NL63, HCoV-OC43, and HCoV-HKU1 (*Zeng et al., 2018*). However, more recently, severe symptoms and fatal outcomes have been caused by three other epidemic viruses, including SARS-CoV, MERS-CoV, and SARS-CoV-2. Coronaviruses express their nonstructural protein in the form of a large protein called polyprotein AB. This polyprotein has to be processed by the host as well as the viral encoded proteases to release approximately 16 NSPs. Two viral proteases share in the digestion of polyprotein AB: the main protease, called 3-C-like protease ($M^{pro}$), and a papain-like protease ($PL^{pro}$) (*Hilgenfeld, 2014*). Both $PL^{pro}$ and $M^{pro}$ have been important targets for drug discovery against SARS CoV, MERS CoV, and SARS CoV-2 (*Kandeel, Altaher & Alnazawi, 2019*; *Li et al., 2019*; *Pillaiyar, Meenakshisundaram & Manickam, 2020*; *Zumla et al., 2016*).

The magic bullet for treating SARS-CoV2 is drug repurposing. As a result, numerous compounds were developed as anti-SARS-CoV-2 agents and therapies or prevent the virus's sequelae, have been studied *in silico*, *in vitro*, and in human clinical trials (*Ghareeb et al., 2021*). Several molecular targets were utilized to develop novel chemicals to combat Coronaviruses such as virus spike (*Choudhary et al., 2021*), main protease (*Mostafa et al., 2021*), papain-like protease (*Delre et al., 2020*; *Mahmoud et al., 2021*), helicase (*Gurung, 2020*) and RNA-dependent RNA polymerase (*Molavi et al., 2021*).

*In silico* drug development investigations focused on the primary protease of SARS-CoV-2. The studies include docking and virtual screening of phytochemicals (*Mandal, Jha & Hazra, 2021*), Hepatic C virus FDA approved drugs (*Uddin et al., 2021*), clinically approved and investigational drugs (*Durdagi et al., 2021*), nonsteroidal anti-inflammatory drugs (*Abo Elmaaty et al., 2021*) and antiallergic agents (*Uras et al., 2021*). A combination of *in silico* and *in vitro* drug repurposing against the SARS-CoV-2 main protease resulted in the identification of several hopeful peptidomimetics (*Zhang et al., 2020*) and small molecules as diclazuril (*Pohl et al., 2021*), nilotinib (*Banerjee et al., 2021*), ritonavir, rotigaptide, and cefotiam (*Durdagi et al., 2021*).

Recently, we provided computational details regarding targeting the $M^{pro}$ and $PL^{pro}$, wherein we used a dataset of FDA-approved drugs (*Kandeel et al., 2020*; *Kandeel & Al-Nazawi, 2020*). In this study, we used a more comprehensive virus-specific and $M^{pro}$-specific dataset of compounds. The selected compounds in this study (Table 1) were selected from chemical libraries of millions of compounds. The compounds were approved by chemical screening compounds libraries providers with the specific aims of (1) conducting a ligand and structure-based search of HTS databases using $M^{pro}$ specific features, (2) performing a 2D fingerprint similarity search against the biologically active compounds from therapeutically relevant viral assays, and (3) exploring specific viral protein binding compounds, antiviral nucleotides, and nucleotide mimetics agents. A total of 37,060 compounds were retrieved and used in our virtual screening, docking, and
**Table 1** The compounds dataset used in this study.

| Library name | Company | Number of compounds | Method of selection |
|---|---|---|---|
| Main protease targeted library | Life chemicals (Niagara-on-the-Lake ON, Canada) | 2,300 | Glide by Schrödinger, SP mode was used to search Life chemicals HTS collection, by using the main protease of SARS-CoV-2 in complex with an inhibitor N3. |
| Antiviral Library by 2D Similarity | Life chemicals | 19,244 | Antiviral Screening Compounds Library was designed with 2D fingerprint similarity search against the 41,514 biologically active compounds from therapeutically relevant viral assays from different virus species. |
| Antiviral Library by Combined Ligand-based and Structure-based Approaches | Life chemicals | 3,500 | Antiviral protein targets were collected from the RCSB PDB. The reference antivirals were collected from ChEMBLdb and clustered according to the target. The top compounds were docked into the target protein and ranked. |
| Antiviral library | Asinex (Winston-Salem, NC USA) | 6,827 | Small molecules and macrocycles with antiviral activity. Specific designs include a-helix mimetics, glycomimetic, diverse synthetic macrocyles, and tri/tetra-substituted scaffolds. |
| Enamine antiviral library | Enamine (Monmouth Jct., NJ, USA) | 4,842 | Nucleoside-like antiviral agents or Nucleoside mimetics from screening collection. The compounds contain natural-like moieties and diverse heterocycles as bioisosters of nucleosides. |
| Antiviral compound library | Selleck (Houston, Texas, United States) | 347 | Collection of antiviral compounds |
| Total no. of compounds | | 37,060 | |

molecular dynamics simulations. The results will help in the design and application of new compounds in treating COVID-19.

## MATERIALS AND METHODS

### Construction of drugs and compounds dataset and ligand preparation

A total of 37,060 compounds dataset was constructed comprising SARS-CoV-2 main protease targeted library, compounds obtained from 2D fingerprint from therapeutically relevant antiviral assays, combined ligand and structure-based approaches of inhibitors of viral proteins (Table 1). All compounds were prepared for virtual screening by Ligprep software using OPLS2005 force field (Files S1 and S2).

### SARS-CoV-2 M$^{pro}$ protein preparation

The structure of M$^{pro}$ (PDB ID 6lu7) was downloaded from the Protein Data Bank. The protein structure was processed and optimized using the Maestro software package's protein preparation wizard (Schrodinger LLC, NY, USA). The protein was protonated, the structure was optimized at cellular pH settings, and the structural energy was minimized using the OPLS2005 force field. The prepared structure was used in all docking and molecular dynamics calculations in this study.

## Virtual screening

Docking of all compounds was performed by Schrodinger Glide docking module. Two-step docking runs were carried out. Initially, the compounds were docked by the standard precision docking protocol (SP docking). Compounds with docking scores of −8.00 or lower were retrieved and subjected to extra precision (XP-docking). This score is suggested to be strong binding compounds with shallow or hydrophobic cavities. The co-crystallized ligand served as the core of a 20-size docking box that encircled the bound ligand in the creation of the docking grid. The obtained results were ranked according to the obtained docking scores.

## Molecular dynamics (MD) simulations

The MD simulations were carried out using GROMACS 5.1.4. (*Abraham et al., 2015*; *Van Der Spoel et al., 2005*). The parameters and optimization of the simulation system were as previously reported (*Al-Hizab & Kandeel, 2021*). Briefly, protein and ligands were handled by AMBERFF14SB and AMBER force field (GAFF). The complexes were dissolved in a single point charge water model in a cubic box of 1.0 nm. For 5000 steps, the solvated M$^{pro}$-ligand complexes were minimized. At 300K, the entire system was equilibrated in two phases: NVT ensemble of 50 ps, followed by NPT ensemble for 1 ns. For all compounds, the production stages were extended to 20 ns. The simulations of the top four compounds were then extended to 100 ns. The pressure and temperature contrls were by Parrinello-Rahman algorithm and V-rescale thermostat algorithm, respectiely. For long-range electrostatics, the Particle Mesh Ewald (PME) technique was utilized (12 Å direct space cut-off). A two fs was chosen as the time step. The output data were collected every 10 ps. In the trajectory analysis, GROMACS MD simulation toolkits were used. The root mean square deviation (RMSD) and per-residue root mean square fluctuation (RMSF) of protein residues were calculated using the g rms and g rmsf functions, respectively. The binding energy was calculated using the g mmpbsa tool (*Kumari et al., 2014*).

## SARS-CoV-2 plaques inhibition assay
### Cell line and virus

African green monkey kidney Vero E6 cells were purchased from the Korean Cell Line Bank (Seoul, Korea). The incubation and handling of cells was as previously described (*Kandeel et al., 2021*). The Korean Cell Line Bank authenticated Vero E6 cells with tests for morphology, growth pattern, histopathology, DNA fingerprinting, and mycoplasma contamination. We also checked the mycoplasma contamination using mycoplasma PCR detection kit (Myco-sniff$^{TM}$ mycoplasma PCR detection kit; MP Biomedicals, Irvine, CA, USA). We prepared stocks for the cell line at early passages, and the cell line was maintained until passage 20 (within 2 months) and then discarded. SARS-CoV-2 S clade (hCoV-19/South Korea/KCDC03/2020, EPI_ISL_407193) was provided by the National Culture Collection for Pathogens (Osong, Korea).

### Virus amplification and virus quantification by plaque assay

Vero E6 cells ($5 \times 10^4$ cells/well 6-well plates) were cultured overnight. The cells were infected with SARS-CoV-2 in PBS (0.1 MOI) for 1 h in a $CO_2$ incubator at 37 °C, then
2 ml of DMEM containing 2% FBS was added. After 3 h incubation in a $CO_2$ incubator at 37 °C, the cells were treated with DMSO (0.1%), birinapant (10 μM) or atazanavir (10 μM) and incubated for 48 h. The virus replication was evaluated using the plaque formation assay. SARS-CoV-2 experiments were approved by the Institutional Biosafety Committee of Hallym University (Permit no. Hallym2020-12) and The amplification of SARS-CoV-2 and the experimental techniques were carried out under a biosafety level 3 (BSL-3) environment.

## Statistical analysis

Correlation statistics were carried out by GraphPad Prism software. Pearson's correlation coefficient was used to conclude the significance of the results.

## RESULTS

### Virtual screening and docking

Recently, we used molecular modeling, virtual screening, and MD) simulation in characterization of the biological aspects of microbial agents, characterization of diseases, and drug discovery (*Altaher & Kandeel, 2016*; *Altaher, Nakanishi & Kandeel, 2015*; *Sheikh et al., 2020*). This study used an antiviral and M$^{pro}$-specific dataset. Virtual screening and docking comprised a two-step process. First, an initial standard-precision (SP) docking protocol was performed, with compounds having a docking score of −8.00 or higher (453 compounds) selected for extra-precision (XP) evaluation. File S1 contains the docked compounds ordered by docking score. After SP-docking, the selected compounds were exported in SDF format and redocked using the XP-docking module, the results of which are provided in File S2. The top 14 compounds with the highest docking scores were used in MD simulations, taking lopinavir as a reference inhibitor (Table 2). All of the top compounds showed favourable profiles and negative scores for Hbond, hydrophobic interactions, vdw, and coulombic interactions. Likewise, the calculated binding energy scores (MM-GBSA) were favourable and indicated strong binding profiles, with values ranging from −56.67 to −106.64 kcal/mol (Table 1).

Statistical analysis comprised determining the correlation between the obtained docking score and ligand efficiency, lipo, Hbond, vdw, coulombic, Glide energy, and binding energy scores (Table 3). A strong negative correlation was observed between docking score and lipophilic interactions ($r = −0.60$, $p > 0.05$), and a positive correlation with columbic interactions ($r = 0.74$, $p > 0.05$). This implies a predominance of electrostatic interactions in compounds binding with SARS-CoV-2 M$^{pro}$.

The determined binding features for each compound with M$^{pro}$ are provided in Fig. 1. The binding site is mostly composed of hydrophobic residues (THR24, THR25, LEU27, VAL42, MET49, PRO52, TYR54, PHE140, LEU141, MET165, LEU167, and THR190); also present are few positively charged residues (ARG188), negatively charged residues (GLU166 and ASP187), and neutral residues (CYS44, SER144, GLN189, and GLN192).

### Molecular dynamics simulations for 20 ns

A potent drug discovery tool is the combination of docking and MD modelling. Drugs can be graded based on their binding affinity and precise interaction with ligand–receptor

**Table 2  Virtual screening and docking output of the top fourteen compounds.**

| Title | Docking score | Glide ligand efficiency | Glide lipo | Glide hbond | Glide evdw | Glide ecoul | Glide energy | MMGBSA _dG_Bind |
|---|---|---|---|---|---|---|---|---|
| Rutin | −11.78 | −0.27 | −2.62 | −0.16 | −47.64 | −27.84 | −75.49 | −88.91 |
| (-)-Epigallocatechin | −11.57 | −0.35 | −2.91 | −0.65 | −34.90 | −24.05 | −58.95 | −70.18 |
| Sennoside A | −10.77 | −0.17 | −2.13 | −0.14 | −39.11 | −18.62 | −57.73 | −61.40 |
| asinex8472 | −9.83 | −0.32 | −2.83 | −1.23 | −38.99 | −11.59 | −50.58 | −64.51 |
| Atazanavir | −9.81 | −0.34 | −3.11 | −1.28 | −37.43 | −13.69 | −51.11 | −74.66 |
| asinex8485 | −9.78 | −0.33 | −2.69 | −1.33 | −42.63 | −11.63 | −54.25 | −73.40 |
| asinex6886 | −9.71 | −0.30 | −3.50 | −0.84 | −45.42 | −8.69 | −54.11 | −56.67 |
| Alpha-Mangostin | −9.14 | −0.31 | −3.68 | −0.83 | −42.12 | −8.79 | −50.91 | −93.46 |
| Glycitin | −8.83 | −0.28 | −2.96 | −0.32 | −32.44 | −15.39 | −47.83 | −77.38 |
| Birinapant | −8.81 | −0.15 | −3.67 | −0.46 | −60.32 | −14.06 | −74.38 | −106.64 |
| F2583-0433 | −8.80 | −0.29 | −2.97 | −1.20 | −42.87 | −17.08 | −59.95 | −89.30 |
| F3234-0818 | −8.65 | −0.30 | −3.03 | −0.90 | −43.91 | −12.30 | −56.20 | −72.79 |
| Lopinavir | −8.68 | −0.15 | −4.90 | −0.16 | −61.64 | −7.72 | −69.36 | −84.25 |
| Cobicistat | −8.55 | −0.10 | −4.48 | −0.26 | −54.69 | −14.25 | −68.94 | −82.33 |

intermediates using these methods. MD simulation and post-dynamic binding energy analysis were performed on the top-ranked compounds from XP-docking. Two stages of compound filtering were used. The RMSD, RMSF, and binding energy values of all 14 compounds were calculated after they were simulated in MD for 20 ns. The top four compounds were studied in a more extensive 100 ns simulation in the second stage. The structural changes in M$^{pro}$ backbone residues were compared (Fig. 2). All treatment complexes, with the exception of Apo M$^{pro}$ and M$^{pro}$ combined with cobicistat and glycitin, showed high stability.

## MM-GBSA binding energies

The MM-GBSA binding energies of the 14 compounds ranged from −42.627 kcal/mol to −42.627 kcal/mol. The top six compounds showed MM-GBSA binding energies ranging from −102.564 to −139.154, indicating a likely substantial binding affinity. Furthermore, all of the investigated compounds had low structural RMSD throughout the 20 ns simulations, with RMSD values as low as 0.21 nm (Table 4).

## Molecular dynamics simulations for 100 ns

To gain a better understanding of the strongest-binding drugs, the four drugs with binding energies greater than −100 kcal/mol (alpha-mangostin, atazanavir, birinapant, and lopinavir) were subjected to 100 ns MD simulations, followed by analyses of RMSD, RMSF, hbond length, and Rg and binding energy. All four had promising binding free energy values (Table 4). Specifically, the estimated MM-GBSA binding energy values were −117.90, −117.83, −121.80, and −112.80 for alpha-mangostin, atazanavir, birinapant, and lopinavir, respectively. The three drugs alpha-mangostin, atazanavir, and birinapant are implied by these values to have superior binding over lopinavir.

Kandeel et al. (2022), *PeerJ*, DOI 10.7717/peerj.12929

**Table 3** Correlation statistics of the obtained docking score and the output parameters of XP-docking.

|  | Docking score *vs.* glide ligand efficiency | Docking score *vs.* glide lipo | Docking score *vs.* glide hbond | Docking score *vs.* glide evdw | Docking score *vs.* glide ecoul | Docking score *vs.* glide energy | Docking score *vs.* MMGBSA_ dG_Bind |
|---|---|---|---|---|---|---|---|
| **Pearson r** |  |  |  |  |  |  |  |
| r | 0.3446 | −0.6011 | −0.1216 | −0.3886 | 0.741 | 0.08881 | −0.3251 |
| 95% confidence interval | −0.2275 to 0.7399 | −0.858 to −0.1035 | −0.6127 to 0.4371 | −0.7621 to 0.1788 | 0.3468 to 0.9127 | −0.4636 to 0.5915 | −0.7298 to 0.2483 |
| R squared | 0.1188 | 0.3613 | 0.0148 | 0.151 | 0.5491 | 0.007888 | 0.1057 |
| ***P* value** |  |  |  |  |  |  |  |
| P (two-tailed) | 0.2275 | 0.0230 | 0.6787 | 0.1697 | 0.0024 | 0.7627 | 0.2567 |
| *P* value summary | ns | * | ns | ns | ** | ns | ns |
| Significant? (alpha = 0.05) | No | Yes | No | No | Yes | No | No |
| Number of XY Pairs | 14 | 14 | 14 | 14 | 14 | 14 | 14 |

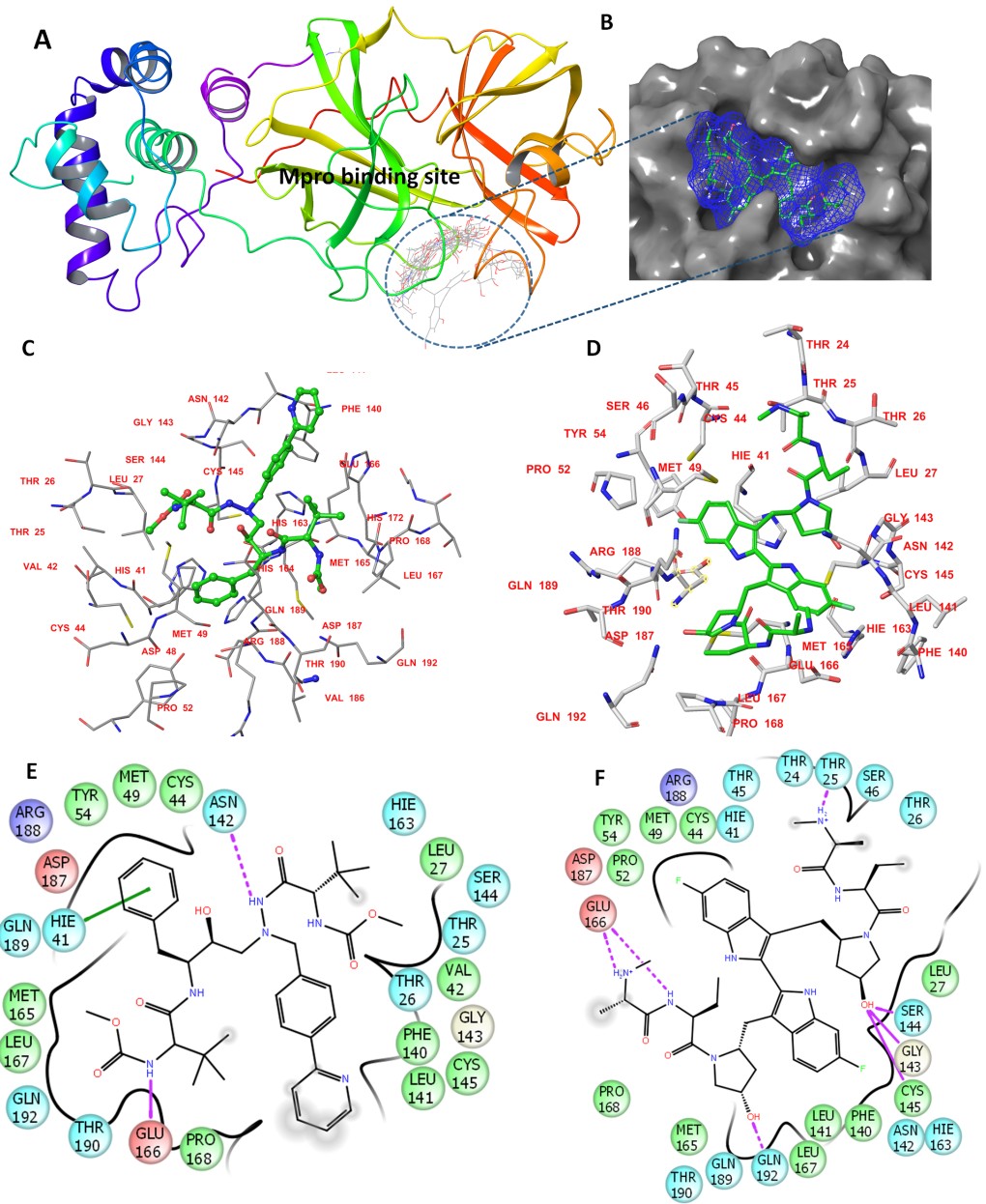

**Figure 1   The docking site and ligands interactions with M^pro.**  (A) The docking site of M^pro following XP docking protocol. (B) Surface representation (blue) of birinapant in the active site of M^pro. (C) The binding site of residues of atazanavir. (D) The binding site residues of birinapant. (E) The ligand interactions of atazanavir. (F) The ligand interactions of birinapant. Hydrogen bonds are shown in purple arrows, hydrophobic interactions in grey circles.

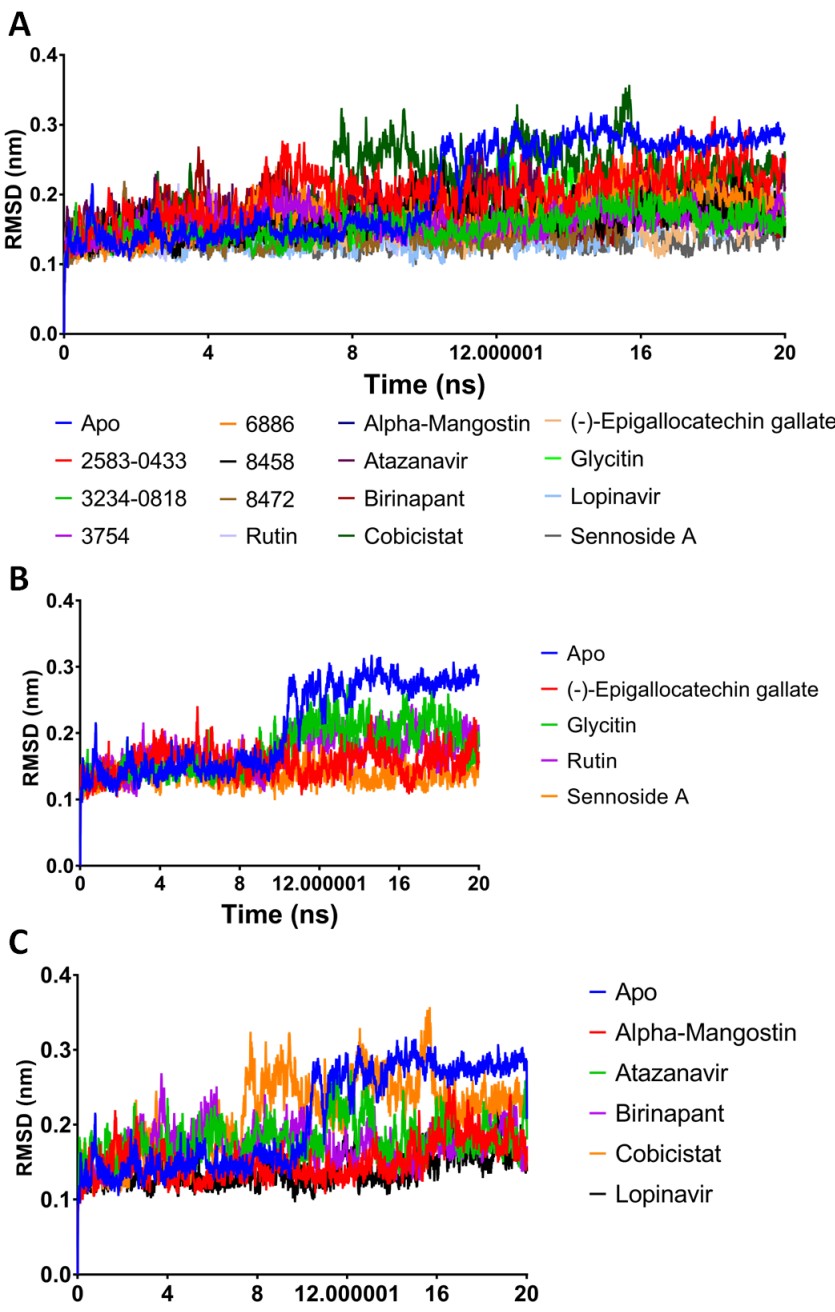

**Figure 2** (A–C) **RMSD plot of the top fourteen compounds after MDS for 20 ns.** Lopinavir was used for reference. Apo structure is M$^{pro}$ without any ligands.

**Table 4** **The MM-GBSA binding energy and the average structure RMSD of the top fourteen compounds after MDS for 20 ns.**

| Compound ID | Binding energy (kcal/mol) | Average structure RMSD (nm) |
| --- | --- | --- |
| Birinapant | −139.154 | 0.171 |
| Atazanavir | −130.299 | 0.180 |
| Lopinavir | −114.654 | 0.138 |
| Cobicistat | −111.296 | 0.214 |
| Alpha-Mangostin | −107.446 | 0.151 |
| 8472 | −102.564 | 0.151 |
| (-)-Epigallocatechin gallate | −88.348 | 0.155 |
| 3754 | −83.21 | 0.160 |
| 3234–0818 | −77.978 | 0.157 |
| 2583–0433 | −73.954 | 0.201 |
| 8458 | −70.288 | 0.165 |
| 6886 | −70.18 | 0.171 |
| Rutin | −47.388 | 0.169 |
| Glycitin | −42.627 | 0.175 |
| Sennoside A | 59.744 | 0.138 |

After 100 ns MDs, average RMSD values of 0.23, 0.20, 0.21, and 0.18 nm were obtained for alpha-mangostin, atazanavir, birinapant, and lopinavir, respectively. Relative to experimental RMSD ranges, these values indicate marked stability of all four drugs when complexed with $M^{pro}$. Such complexes can be ranked in terms of stability as follows: lopinavir>atazanavir>birinapant>alpha-mangostin. The low ranking of alpha-mangostin can be explained by the abrupt drift in its RMSD value at 22 nm, observable in Fig. 3. The energy value obtained for alpha-mangostin likewise indicates a lower affinity to $M^{pro}$. Meanwhile, the per-residue RMSF (Fig. 4) shows conserved RMSF features in $M^{pro}$ complexes with lopinavir, birinapant, and atazanavir. Surprisingly, alpha-mangostin showed several protein fragments with very high RMSD values of 0.4 nm. Nonetheless, based on observations of binding energy, RMSD, and RMSF values, we can exclude alpha-mangostin from being repurposed on the basis of interaction with SARS-CoV-2 $M^{pro}$.

## Radius of gyration

The radius of gyration can be used to determine the compactness of a system, with lower Rg values indicating more stable structures and higher Rg values indicating less compactness or more unfolded protein. All four top drugs had an average Rg value of 2.21 nm; these similar Rg values indicate the stability of the examined drugs when complexed with $M^{pro}$. Figure 5 shows the variation in Rg obtained during 100 ns MD simulations. Birinapant and ritonavir showed almost similar profiles with less-variable Rg, while alpha-mangostin

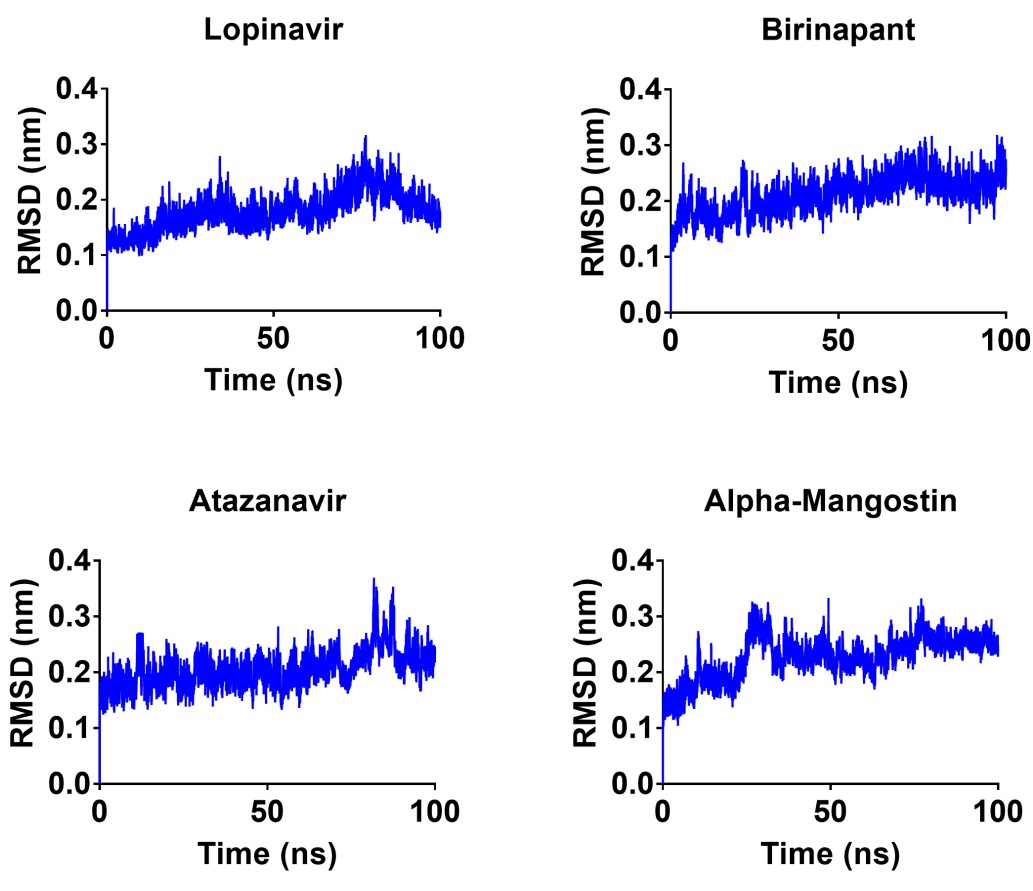

**Figure 3**  RMSD plot of the top four compounds, alpha-mangostin, atazanavir, birinapant and lopinavir, after MDs for 100 ns.

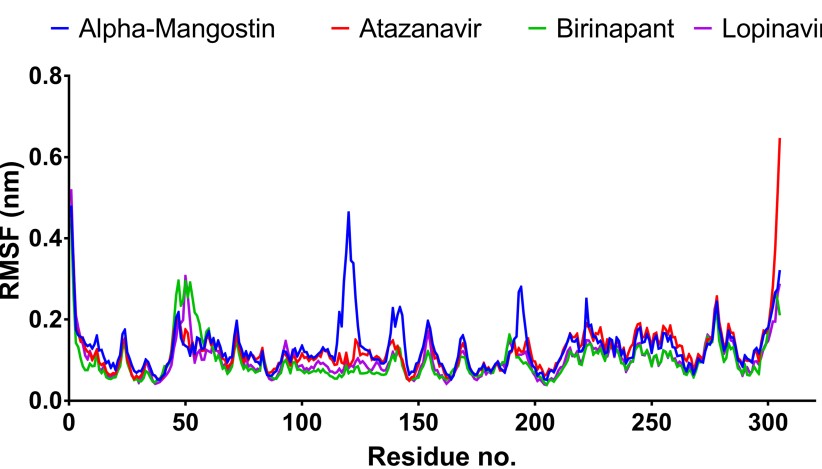

**Figure 4**  RMSF plot of the top four compounds after MDs for 100 ns.

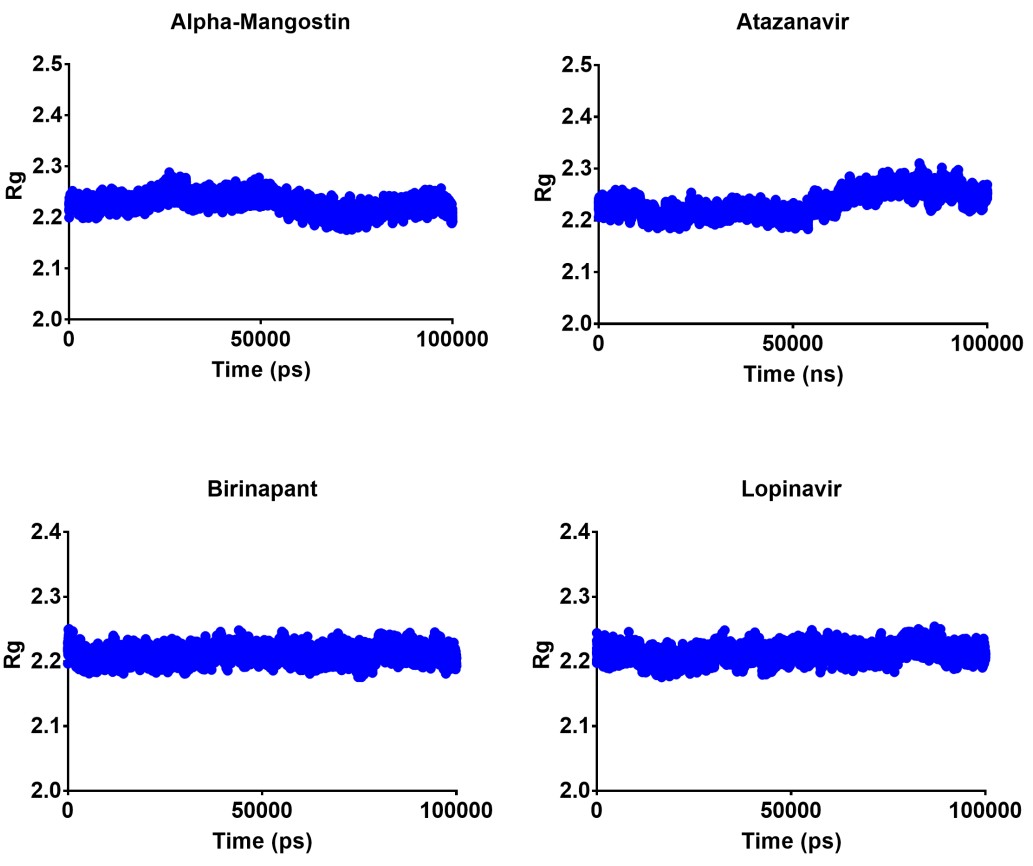

**Figure 5    Radius of gyration of the top four compounds after MDs for 100 ns.**

and atazanavir showed biphasic profiles of alternating higher and lower Rg. Nonetheless, the overall average Rg values were similar for the four drugs.

## Hydrogen bond length

Figure 6 shows the average hydrogen bond length obtained using GLU166 over a 100 ns simulation. Birinapant, with an average length of 0.25 nm, demonstrated the only stable binding with GLU166.

## Decomposition of MM-GBSA binding energy

The primary interactions during drug recognition by M$^{pro}$ were studied using post-dynamic energy decomposition analysis (Table 5). The findings revealed that vdw and electrostatic interactions were the most critical forces for all four drugs. More specifically, vdw was the major force for alpha-mangostin, atazanavir, and lopinavir, while electrostatic forces were the major contributor for birinapant binding, with a lesser contribution from vdw.

## SARS-CoV-2 Plaque inhibition assay

Plaques inhibition assays in Vero E6 cells were used to explore the drug inhibitory properties against SARS-CoV-2 infection. At a concentration of 10 μM, atazanavir had no antiviral

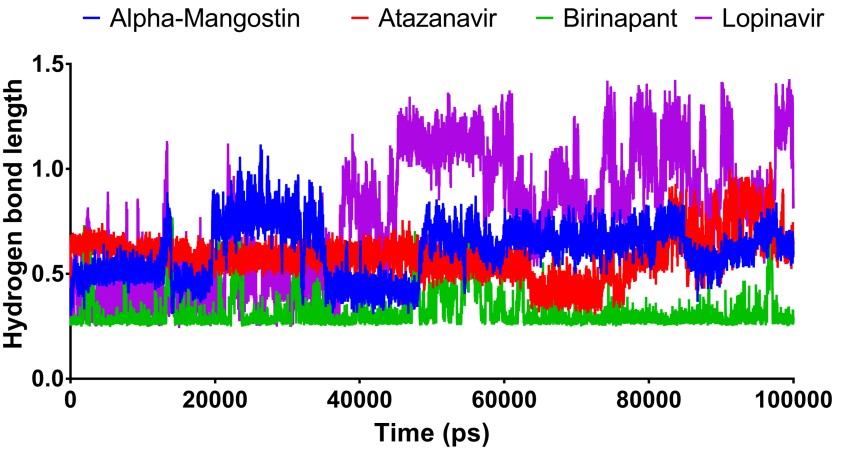

**Figure 6** The hydrogen bond length of the top four compounds after MDs for 100 ns.

**Table 5** Decomposition of the estimated MMGBSA binding energy for the binding of alpha-mangostin, atazanavir, birinapant and lopinavir with SARS-CoV-2 M^pro.

|  | Alpha-Mangostin | Atazanavir | Birinapant | Lopinavir |
|---|---|---|---|---|
| van der Waal energy | −184.419 | −292.82 | −134.625 | −227.389 |
| Electrostattic energy | −28.745 | −81.919 | −305.47 | −40.001 |
| Polar solvation energy | 113.399 | 287.534 | 335.31 | 177.784 |
| SASA energy | −18.036 | −30.595 | −16.679 | −23.173 |
| Binding energy | −117.863 | −117.827 | −121.346 | −112.801 |

effects. Birinapant, on the other hand, reduced the production of SARS-CoV-2 plaques by 37% (Fig. 7). Treatment with birinapant significantly inhibited the SARS-CoV-2 plaque formation in a dose-dependent manner. The estimated $IC_{50}$ values for birinapant was 18 ±3.6 µM.

## DISCUSSION

With the emergence of SARS-CoV-2 in December 2019 and its rapid worldwide spread, drug repurposing has been one tool available to combat the disease. Many drugs with proven efficiency and safety have been repurposed for other clinical applications. Sildenafil is one such drug; it was initially produced to treat angina and later used for male erectile dysfunction (*Goldstein et al., 1998*); zidovudine was repurposed earlier from an anticancer drug to an anti-HIV agent (*Ashburn & Thor, 2004*), and the antidepressant dapoxetine has been effective in managing premature ejaculation (*Fu, Peng & Hu, 2019*).

The SARS-CoV-2 M^pro has been an attractive target in many drug discovery studies. The M^pro was targeted by a number of compound libraries, including those containing drugs approved by the Food and Drug Administration (*Kandeel & Al-Nazawi, 2020*), flavonoids and natural compounds (*Joshi et al., 2020*; *Vijayakumar et al., 2020*), tetracycline (*Bharadwaj et al., 2020*) and microbial natural products (*Sayed et al., 2020*). In this study,

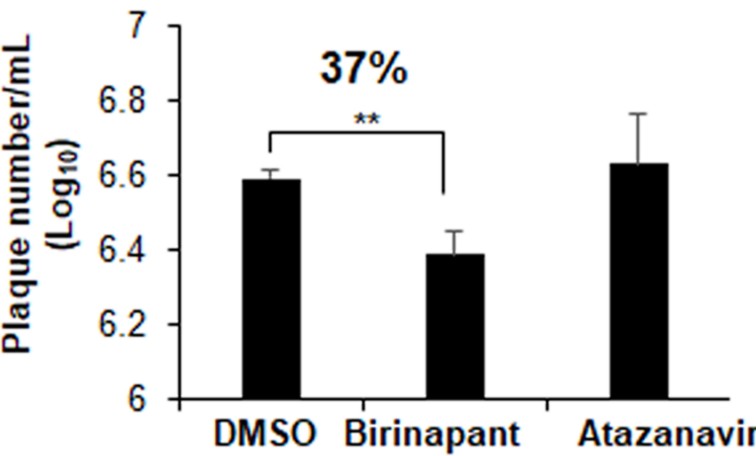

**Figure 7** **Effect of birinapant and atazanavir on the replication of SARS-CoV-2.** Vero E6 cells were infected with 0.1 MOI SARS-CoV-2 in 6-well plate and then treated with DMSO (0.1%), birinapant (10 µM) or atazanavir (10 µM) at 3 h after virus infection ($n = 3$). Supernatants of virus-infected cell cultures were collected at 48 h after virus infection. Virus replication in the supernatants was quantified by plaque formation assay. **$p < 0.01$.

a large library of virus protein-specific compounds was selected. Initial docking showed interesting docking scores and favourable profiles of structure stability and binding energy. The four drugs selected for the final comprehensive 100 ns MD) simulations were alpha-mangostin, atazanavir, birinapant and lopinavir.

Lopinavir is a well-known protease inhibitor with high efficiency against the HIV-1 protease (*Cvetkovic & Goa, 2003*). In addition, the drug was effective against SARS-CoV and MERS-CoV (*Chan et al., 2015*) and improved the health markers in SARS-CoV-2–infected patients (*Ye et al., 2020*). Atazanavir is another HIV-1 protease inhibitor with the advantage of oral administration in combination with other antiretroviral drugs (*Goldsmith & Perry, 2003*). Birinapant is an apoptosis inhibitor, and it has approved efficiency in controlling viral hepatitis in combination with other antiviral drugs (*Testoni, Durantel & Zoulim, 2017*). Previous *in silico* research showed that birinapant could bind to the main protease of the SARS-CoV-2 virus (*Lokhande et al., 2021*). In our study, in comparison with lopinavir, both atazanavir and birinapant showed an improved XP-docking score, higher binding energy and a lower structural root-mean-square deviation (RMSD) during 100 ns MD simulations. Therefore, based on the clinical efficiency of lopinavir against SARS-CoV-2, the drugs atazanavir and birinapant are expected to perform with similar or improved efficacy comparable to that of ritonavir. Interestingly, it was recently shown that atazanavir can inhibit M$^{pro}$ activity while simultaneously suppressing SARS-CoV-2 replication (*Fintelman-Rodrigues et al., 2020*).

The strong binding profiles of atazanavir and birinapant are supported with hydrophobic interactions. In addition, atazanavir formed two hydrogen bonds with ASN142 and GLU166, and birinapant formed four hydrogen bonds with THR25, GLU166, and GLN192 and a tridentate bond with GLY143, SER144 and CYS145 (Figs. 1E and 1F).

In the second rank, following atazanavir and birinapant, alpha-mangostin and cobicistat showed quite high binding energy but had less structural stability owing to higher RMSD values. Cobicistat is a cytochrome enzyme inhibitor used to increase the systemic availability of other antiviral drugs, such as atazanavir (*Xu et al., 2010*).

After screening investigations, the top two compounds were used in antiviral assays. Atazanavir had no antiviral effects. Atazanavir has been shown to have anti-SARS-CoV-2 action in previous studies (*Fintelman-Rodrigues et al., 2020*). The lack of antiviral effectiveness of atazanavir found in our investigation might indicate that slight differences in the type of virus utilised could alter atazanavir efficacy. In contrast, birinapant has antiviral properties at low micromolar concentrations. The measured EC50 value of birinapant (18 µM) coincides with the measured value for other known antiviral drugs such as remdesivir and lopinavir, which yielded EC50 values against SARS-CoV-2 replication at 23.15 and 26.63 µM, respectively (*Choy et al., 2020*). Birinapant's anti-SARS-CoV-2 activity supports further investigation into its usage as an anti-COVID-19 medication. Combining birinapant with other antivirals may result in considerable SARS-CoV-2 virus particle elimination.

## CONCLUSION

After a comprehensive study involving virtual screening, docking, and MD simulations of a unique set of antiviral agents, two highly potent $M^{pro}$-binding drugs, birinapant and atazanavir, showed promise. These drugs had improved energetic and structural stability profiles that were comparable to or higher than those produced by the classic antiviral protease inhibitor ritonavir. Birinapant was found to inhibit SARS-CoV-2 replication with promising inhibition in the low micromolar range.

## ACKNOWLEDGEMENTS

The authors acknowledge King Faisal University for providing facilities and labs.

### Funding

This project is funded by the Ministry of Health, Saudi Arabia, Project number (495) and date 11/9/1441H. Mahmoud Kandeel received financial support from the Ministry of Health, Saudi Arabia for searching for new binding drugs with SARS-CoV-2 main protease. The funders had no role in study design, data collection and analysis, decision to publish, or preparation of the manuscript.

### Grant Disclosures

The following grant information was disclosed by the authors:
Ministry of Health, Saudi Arabia: 495.

### Competing Interests

The authors declare there are no competing interests.

## Author Contributions

- Mahmoud Kandeel conceived and designed the experiments, performed the experiments, analyzed the data, prepared figures and/or tables, authored or reviewed drafts of the paper, and approved the final draft.
- Jinsoo Kim performed the experiments, analyzed the data, prepared figures and/or tables, authored or reviewed drafts of the paper, and approved the final draft.
- Mahmoud Fayez and Yukio Kitade analyzed the data, authored or reviewed drafts of the paper, and approved the final draft.
- Hyung-Joo Kwon conceived and designed the experiments, analyzed the data, authored or reviewed drafts of the paper, and approved the final draft.

## Data Availability

The screening results and PDB files are available in the Supplementary Files.

## Supplemental Information

Supplemental information for this article can be found online at http://dx.doi.org/10.7717/peerj.12929#supplemental-information.

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
