# Peer review of "Antiviral drug discovery by targeting the SARS-CoV-2 polyprotein processing by inhibition of the main protease"

_PeerJ, doi:10.7717/peerj.12929_

## Round 0.1 · original submission · Major Revisions

All the reviewers recognise the potential worth of this study, but two in particular express concerns that (i) you have not cited other relevant work with sufficient depth and clarity, and (ii) that some of the work needs more detail in description. Their comments are clearly described in their report and I won't reiterate them here.

Please address all comments in a rebuttal letter and in your revised manuscript.

·

Basic reporting

The manuscript is written very well. The concepts are clearly defined.
English language of the manuscript is fine.
Introduction and background are good enough to present the work of the manuscript.

Experimental design

The authors have given sufficient details of the methodology in the manuscript.
Methods can be repeated to follow the results.

Validity of the findings

Research of this manuscript was done computationally, and authors have supported theirs finding by experimental work which is a strength of this manuscript.

Additional comments

GENERAL COMMENTS
The following are my comments which are needed to be solved before publication.
Abstract-
1-Line 20.
“To date, no new drug or vaccine has been approved for treating SARS
CoV-2 infections”
Remove “Vaccine”
2-Line-72-73. How did the authors collect a total of 37060 compounds dataset, write in detail so readers can follow your work?
3-Line 86
Docking of all compounds was by Schrodinger glide docking module.
Correct above sentence by adding “verb”.
4-Line 88.
“Compounds with docking score of -8.00 or higher were retrieved”
What was basis to choose “ -8.00 or higher”?
5- In Line 89 -90, Add x y and z coordinates.
6-Line 100, correct 50ps , it should be 50 ps. Check for such errors.
7-Line 114 , add reference for “g_mmpbsa tool”.
8- Figure 1 , A. B, C, and D remove the black background. Improve picture quality.
9- Figure 5, Correct scale of X-axis ( It should be maximum 100 ns but it is 100000 ns)
10- Since authors have performed experiment by using SARS-CoV-2 virus. Add an approval of bioethics statement to conduct the experiment by using live virus.

Reviewer 2 ·

Basic reporting

The presented work seeks to discover antiviral compounds which may be candidates for treating COVID-19. The question of which drug treatments might improve patient outcomes in COVID-19 is the subject of ongoing research in the field. The presented work follows a valid investigation and contributes a result towards that research effort.

The article is written with clear English. There are some minor mistakes with spelling and grammar which do not substantially affect the article's content. There is an issue with the capitalisation of some proper nouns in the article, such as "glide" (referring to the docking software "Glide"), "protein data bank" ("Protein Data Bank"), and so on. On lines 102-103, the sentence "production stages were done over simulation times of 20 and 100 ns with NPT ensemble was adopted" is confusing, and on line 104 "algorithmfor" is missing a space. There should be a space between numbers and their units, such as in "50ps" ("50 ps") or "300K" ("300 K"). Although these are minor issues, the authors should consider double-checking the prose in order to improve the readability of the article.

The introduction adequately describes the motivation for the research. However, the section does not support or contextualise the in vitro screening part of the study, despite this being crucial to its core output, which is the 37% reduction of replication after 48 hours demonstrated for Birinapant. The authors should expand the introduction to the article to make clear the importance of the in vitro assay to produce evidence for establishing a candidate treatment. The Introduction section also lacks a sufficient review of similar in silico / in vitro screening studies aimed at drug / antiviral repurposing, so such a review should be undertaken and added.

Experimental design

The authors have chosen to undertake in silico screening first, followed by an in vitro screen for the best-scoring compounds, and this is a standard approach for drug repurposing. The two stages of docking and the stage of molecular dynamics have been described well.

Neither the section "Construction of drugs and compounds dataset" beginning line 71, nor the referenced Table 1, describes the fingerprint or structure similarity methods used to create the compound library. This section must be made more detailed. There are three questions which should be answered: what were the seed compounds of the search (known antivirals), how were compounds generated to be tested against those seeds, and which methods were used to assess compound similarity?

On line 72 and in Table 1, it is written that a set of 37,060 compounds were tested, but in Supplementary File 1 ("SP docking results") 47,068 results are listed. No subsequent references to Supplementary File 1 explain the discrepancy. The authors should describe more clearly the dataset that is being presented in Supplementary File 1. Which 47,068 compounds were used?

Vero E6 is an appropriate cell line for this kind of SARS-CoV-2 infection study. The provenance is described as the Korean Cell Line Bank in the subsection "Cell line and virus" beginning on line 116. The authors' description is missing an account of how the cell line has been authenticated and checked by the authors for possible contamination, which should be part of the quality control process for this experiment [DOI:10.1023/A:1022949730029, DOI:10.1038/bjc.2014.166]. The same subsection outlines that the SARS-CoV-2 sample used by the authors was NCCP No. 43326. The authors may wish to insert a sentence to clarify that viral samples based on NCCP No. 43326 are the wildtype virus, i.e. GISAID clade S, PANGOLIN lineage A. Variants of SARS-CoV-2 are of pressing importance, and the wildtype is no longer the most relevant strain to the COVID-19 pandemic, so explicitly mentioning the lineage used in the experiment will help clarify the applicability of the research.

Validity of the findings

The in silico portion supports prior in silico investigations regarding the antiviral drug Birinapant, which conclude it has the potential to limit the replication of SARS-CoV-2 by inhibiting Mpro [DOI:10.1080/07391102.2020.1805019]. The in silico work also highlights Atazanavir, for which there are also ongoing investigations [DOI:10.1101/2020.04.04.020925, DOI:10.1186/s13063-020-04987-8]. The in vitro validation focuses on these two drugs and presents a positive result for Birinapant inhibiting the replication of SARS-CoV-2 in the Vero E6 cell line at 10 μM after 48 hours.

In lines 19-20 of the abstract, a claim is made that "no new drug or vaccine has been approved for treating SARS CoV-2 infections". The antiviral drug Remdesivir has already been approved for the treatment of COVID-19 (e.g. by the MHRA in the UK), which followed the results of the ACTT-1 trial last year [DOI:10.1056/NEJMoa2007764]. Monoclonal antibodies have also recently been approved (e.g. by the MHRA) as a treatment for COVID-19 [DOI:10.1136/bmj.n2083]. There are also now clearly many approved vaccines which are efficacious at preventing COVID-19. The statement appears to be severely outdated and must be updated to reflect the current state of research into treatments for COVID-19.

In line 36 of the abstract, it is claimed that "Birinapant can be used to treat COVID-19." This is a strong claim which is not adequately supported by the experimental result presented, which is a 37% reduction of replication in an in vitro assay based on the Vero E6 cell line. Claims that something is a treatment should be supported by clinical evidence. In the case of this research, it would be more accurate to conclude that Birinapant is a candidate for further investigation in order to establish whether it is an effective treatment. A similar point is made in the Discussion section on lines 270-271, saying "Birinapant's anti-SARS-CoV-2 activity supports its usage as an anti-COVID-19 medication", and this should also be updated to say e.g. "...supports further investigation into its usage...".

The results of the docking and molecular dynamics stages of the study are well-presented and sufficient raw datasets for those stages have been provided in the supplementary information.

The caption for Figure 7 lists the concentration tested as 10 mM, which is also present in lines 128-129 of the Methods section. However, a value of 10 μM is written in the abstract as well as in the "SARS-CoV-2 Plaque inhibition assay" subsection of the Results section. The value tested was presumably 10 μM, as per similar studies, so the authors should clarify or correct the "10 mM" values. This subsection refers to "Fig. 6", but it seems the matching figure would be "Fig. 7". Furthermore, it would be helpful for the authors to supply a table of raw data for Figure 7 in the supplementary information.

On line 230, it is stated that "the estimated IC 50 values for birinapant was 18±3.6 µM." It must be stated, possibly in the Methods section, how this value is estimated, and any further raw data that is relevant should be provided in a figure in the main text and in a table in the supplementary information. Can the authors produce a dose-response curve to support the asserted value? The value should then be related to comparable values in other in vitro COVID-19 replication inhibition studies, such as to Remdesivir in Vero E6 cells [DOI:10.1016/j.antiviral.2020.104786].

The Discussion section does not mention important context from the field for the two compounds that were primarily investigated, Birinapant and Atazanavir. Birinapant was highlighted last year by an in silico study following a similar method of docking then molecular dynamics [DOI:10.1080/07391102.2020.1805019]. The authors should relate their work to this earlier study - the in silico part supports its conclusions, and the in vitro work extends the evidence for the candidacy of Birinapant for further work. Atazanavir has also been the subject of a mixed in silico / in vitro investigation [DOI:10.1128/AAC.00825-20]. Like in this work, it was found by Fintelman-Rodrigues et al. that Atazanavir undergoes stable binding with a strong binding energy to Mpro. However, the Fintelman-Rodrigues study found that Atazanavir exhibits antiviral activity in Vero E6 cells in vitro, and then goes on to establish that the inhibition of replication holds in human cells. The disagreement between this work and the in vitro part of the Fintelman-Rodrigues study should be discussed, particularly in the context of upcoming clinical trials involving Atazanavir [DOI:10.1186/s13063-020-04987-8].

Additional comments

The core research behind this work, in which Birinapant is selected in silico from a large panel of compounds and then its activity verified in vitro, is promising. However, the article would need major revision in order to fully describe the work done and to contextualise its place within the current body of literature. Particularly, there are methods used which have not been described; the authors have not related the work to any relevant developments in COVID-19 therapeutics over the last year, including comparable in silico and in vitro studies; and some of the stated discussion and conclusions of the study are not supported by the results generated.

Reviewer 3 ·

Basic reporting

“To date, no new drug or vaccine has been approved for treating SARS CoV-2 infections” – In fact there are many vaccines that have been approved and a host of new drugs are in development. This is clearly a naive statement that should be revised.

Generally, the authors do not acknowledge the work of other groups in this area. For example, birinapant has been identified by other studies of a similar nature to that submitted by the authors but these are not cited.

e.g.
https://www.tandfonline.com/doi/full/10.1080/07391102.2020.1805019
https://www.tandfonline.com/doi/full/10.1080/07391102.2020.1768151

I feel that many of the compounds identified in this study have been identified by others with a similar or identical approach - the authors do little to recognise this.

Experimental design

The experimental design appears to be sound and the plaque assays are an important addition.

Validity of the findings

The paper is clearly written and the findings of the paper are generally supported by the data presented.
RMSD and gyration figures are very difficult interpret and should be separated. E.g. Fig. 5. the data for birinapant is barely visible.

Additional comments

Generally, this is a useful addition to the SARS-CoV-2 molecular docking repurposing genre, especially since the wet lab data is included. I feel that many of the compounds identified in this study have been identified by others with a similar or identical approach - the authors do little to recognise this.

---

## Round 0.2 · accepted · Accept

I apologise for the length of time it has taken to deal with this resubmission. I am satisfied that all points raised have been addressed and am delighted to recommend acceptance.